# Supplemental C Addressed the pH Conundrum in Sustainable Marine Aquaponic Food Production Systems

**DOI:** 10.3390/foods12010069

**Published:** 2022-12-23

**Authors:** Yu-Ting Chu, Yiwen Bao, Jen-Yi Huang, Hye-Ji Kim, Paul B. Brown

**Affiliations:** 1Department of Forestry and Natural Resources, Purdue University, West Lafayette, IN 47907, USA; 2Department of Food Science, Purdue University, West Lafayette, IN 47907, USA; 3Environmental and Ecological Engineering, Purdue University, West Lafayette, IN 47907, USA; 4Department of Horticulture and Landscape Architecture, Purdue University, West Lafayette, IN 47907, USA

**Keywords:** sustainable food production, pH, supplemental C, marine aquaponics, wastewater treatment, halophytic plants, *Litopenaeus vannamei*

## Abstract

pH is the major issue that concerns all producers in aquaponics, as the main three organisms (aquatic animal, plant, and microbes) have different preferences. Additional C is a potential approach to amend the growing environment and improve shrimp and plant growth, and microbe establishment. Aquaponics under saline conditions has, however, not been studied in detail in regard to the effect of pH and additional C. In this study, we evaluate the impact of pH and additional C on the growth of Pacific whiteleg shrimp and five edible plants (three halophytes and two glycophytes) in marine aquaponic systems using nutrient film technique (NFT). The results indicated that plants grow better in both pH 6.5 treatments; however, additional C improved the growth in pH 7.5 + C treatment and had similar yield to lower pH treatments. The results indicated both pH and additional C had little impact on shrimp growth. In conclusion, adding C can be a practical solution to the pH conundrum for marine aquaponics. Appling additional C was suggested for the operation of marine aquaponic food production system when the pH is high.

## 1. Introduction

The demand for sustainable food production systems, and awareness of food safety has increased, driven by an increasing global population, urbanization, changing food habits, climate change, resource depletion, and environmental issues [1,2,3]. Aquaponic food production systems produce more food with less environmental impact than traditional agriculture and use less freshwater and land [4,5,6]. Moreover, in many parts of the world, these systems must be in closed areas (buildings), providing the additional benefit of biosecurity and reduced pesticide and herbicide usage [4]. It has been considered a sustainable food production system and a solution to food safety and security [2,4,6]. Furthermore, incorporating seawater or brackish water systems offers the potential of conserving freshwater resources, increasing food diversity, offsetting the burden of declining seafood production from the wild supply [2,7,8], and may become an important component of our global food production complex in the future.

Nutrient flows in aquaponics vary as a function of feeding frequency and regime, feed amounts, animal to plant ratio, hydraulic loading rate, subsystem ratio, and pH [9,10,11,12,13,14,15,16]. Environmental pH plays a critical role in nearly all chemical reactions in water, and influences physiological functioning of aquatic animals, plants, and microorganisms [4,13,17,18,19,20]. One of the major challenges in freshwater aquaponic systems is the variation in optimal pH for the three taxa; animal, plant and bacteria (i.e., the pH conundrum). The pH optimum for aquatic animals and bacteria differs from the optimum for plants [4]. However, Gunning et al. [8] reported that the ideal pH for saltwater aquaponics was 7.5 to 8.5, as halophytes originated from coastal areas with high alkalinity and pH. Therefore, the probability of a pH conundrum in marine or brackish water aquaponic systems may be less than in freshwater systems. Nevertheless, Chu and Brown [9,21] mentioned that pH was a possible factor influencing growth of the halophytes they evaluated. The ideal range of pH for an integrated system that includes aquatic animals, plants and bacteria in marine aquaponics remains largely unknown and may significantly influence compatibility of biotic subsystems and significantly influence outputs.

Adding molasses is a common approach in field agriculture to amend the growing environment for better growth and yield [22,23]. Chu and Brown [9] found halophytic plants grew better with higher C/N ratio in deep-water culture systems. However, there is little research on the effect of additional C on plants grown at different pH levels. On the other hand, adding organic carbon is the most common practical approach used in sustainable aquaculture to stimulate the formation of biofloc [24,25]. Biofloc includes heterotrophic and autotrophic bacteria, algae, zooplankton, and various microorganisms in water to control and manage toxic nitrogen compounds [24,25]. Additionally, biofloc suspended in the water column can serve as a nutritional supplement for aquatic animals that can tolerate high turbidity. Recommended C/N ratio is 10 to 20 [26,27,28]. The hypothesis of this research is supplemental C can address the pH conundrum in marine aquaponics. We tested by evaluating the effect of pH levels and additional C on the growth and production of whiteleg shrimp and five plant species in experimental marine aquaponic systems.

Pacific whiteleg shrimp (*Litopenaeus vannamei*) accounts for 52.9% of total crustacean production and was the second largest aquaculture industry in 2018 [29]. Rapid growth, good survival in intensive culture, wide range of salinity tolerance, high market price, and increasing demand for whiteleg shrimp attract farmers’ attention in the industry. Compared to fish-based aquaponics, which relies on plants as the major revenue, shrimp-based aquaponics, in which shrimp can be harvested 3 to 4 times per year, has the possibility of alleviating the economic drain caused by one harvest of fish per year [30]. Moreover, shrimp are one of the species that can benefit from the addition of C to stimulate the formation of biofloc.

Three halophytic plants, red orache (*Atriplex hortensis*), okahijiki (*Salsola komarovii*), and minutina (*Plantago cornonpus*) are nutritious edible plants that have been evaluated in marine aquaponics [31,32,33,34]. Further, glycophytic plants that can tolerate salt have the potential to be grown in saltwater systems and help the development of marine aquaponics. Due to high nutritional value and delicious taste, Swiss chard (*Beta vulgaris*), and kale (*Brassica oleracea*) are the most commonly grown vegetables worldwide [35,36]. Moreover, Grieve et al. [37] reported that they are able to tolerate saline environments, which suggests that they are potential crops in marine aquaponics.

## 2. Materials and Methods

### 2.1. Aquaponic System Design

Twelve nutrient film technique (NFT) aquaponic systems were constructed in the greenhouses at Purdue University, West Lafayette, IN, USA. Each aquaponic system was assembled with a 64.4 L aquaculture tank, three plant growing channels, and an 18.9 L biofilter tank (Figure 1). The aquaponic systems were filled with reverse osmosis (RO) water, and sea salt (Instant Ocean^®^, Blacksburg, VA, USA) to adjust the salinity to 15 ppt [31]. On top of aquaculture tanks, lids and plastic mesh were applied to prevent shrimp from escape. Biofilter tanks were equipped with filter bags filled with 6 L of bio-media (surface area 274 ft^2^/ft^3^; Pentair Aquatic Eco-Systems, Inc., Apopka, FL, USA). Dissolved oxygen (DO) concentration was maintained above 6 mg/L via aerating through air stones installed in every aquaculture tank and biofilter tank. Water temperature was maintained within the range of 26 to 30 °C for the shrimp using submersible heaters (300 w; Aqueon, WI, USA) installed in aquaculture tanks. Airlifts were used to promote water flow in the systems at a rate of 3 L/min.

### 2.2. Biological Material

#### 2.2.1. Shrimp

Juvenile Pacific white shrimp (*Litopenaeus vannamei*) were provided by a commercial shrimp farm (RDM Aquaculture, Fowler, IN, USA) and transported to the greenhouse. Transport conditions included water temperature of 24 °C, salinity 15 ppt, and pH 8.2. Shrimp were separated into two 700 L tanks and quarantined for a week before moving into experimental units. Shrimp were fed a commercial shrimp feed (Zeigler Brothers, Gardners, PA, USA) twice daily (8:00 a.m. and 5:00 p.m.) during the quarantine period. Daily feedings were calculated based on 3.0% of total biomass and divided into equal aliquots. To assure that all shrimp in all treatments undergo with same unit pH changes on the day of transferring into experimental systems, pH in the two 700 L tanks was adjusted with 10% H_2_SO_4_ at a rate of 0.5 units per day and maintained the level when pH 7.0 was reached during the quarantine. The pH value in experimental systems was adjusted to desired levels 6.5 or 7.5 before the transfer.

#### 2.2.2. Plant Material and Germination Conditions

Seeds of three halophytic plants, red orache (*Atriplex hortensis*), okahijiki (*Salsola komarovii*), and minutina (*Plantago cornonpus*), and two glycophytic plants, Swiss chard (*Beta vulgaris*), and kale (*Brassica oleracea*) were purchased from a commercial source (Johnny’s Selected Seeds, Winslow, ME, USA). Seeds were sowed in horticubes, soilless foam medium (OASIS^®^ Grower Solutions, Kent, OH, USA) and held in trays. During the germination stage, seeds were maintained in a growth room in the greenhouse and irrigated with fresh water for the first week. Starting at the beginning of the second week, salinity in the irrigation water was gradually increased by adding sea salt (Instant Ocean^®^, Blacksburg, VA, USA) at a rate of 2 to 3 ppt every 48 h to alleviate osmotic shock on seedlings. Salinity was maintained when the desired salinity, 15 ppt, was reached. Heating pads (Vivosun^®^, Redding, CA, USA) were used under trays to maintain the water temperature at 22 °C. The room temperature in the growth room was set at 20 ± 1 °C and provided with 16 h photoperiod at 150 μmol m^−2^ s^−1^ photosynthetically active radiation (PAR).

### 2.3. Experimental Design and System Management

The experiment was a 2^2^ full factorial, two factors each with two levels, 4 combinations in triplicate for a total of 12 aquaponic systems. The two factors evaluated were pH and additional carbon. The levels of pH were 6.5 and 7.5. The levels of additional carbon were with or without supplementation. Treatments were designated 6.5 + C, 6.5 No C, 7.5 + C, and 7.5 No C. For the treatments with additional carbon, molasses (Hawthorne Gardening Co., Vancouver, WA, USA) was provided as an organic carbon source for the adjustment of the C/N ratio. The amount of molasses added was determined on the carbon-nitrogen content of shrimp feed and the carbon content of the molasses to raise the C/N ratio to 10. Molasses and shrimp feed samples were ground and filtered through a 10-mesh sieve. 30.0 mg of sample material was placed into an empty sample tin, which was wrapped into a ball and analyzed in the Department of Agronomy, Purdue University using the FlashEA (C/N machine, Swedesboro, NJ, USA). A week before the experiment started, shrimp were weighed and acclimated in aquaculture tanks to generate nutrients for plants. In order to reduce the stress, we measured 7 shrimp at a time and measured 4 times to get 28 shrimp per tank instead of measuring individually. The stocking density of shrimp was 127 shrimp/m^2^ (28 shrimp/tank; 451 shrimp/m^3^; 1.0 kg/m^3^), the average size of shrimp was 2.25 g per individual. A total of 18 plants (6 plants per species) were stocked in the three plant growing channels, which was equivalent to a density of 23 plants/m^2^. Initial fresh weights were 0.95, 0.38, 0.34, 0.81, and 1.19 g for red orache, minutina, okahijiki, Swiss chard, and kale, respectively. Shrimp were provided a total feed allotment of 3% of their body weight in three separate aliquots each day at 8 am, 12 pm and 5 pm. A commercial shrimp feed (Zeigler Brothers, Gardners, PA, USA) with 40% protein, 9% fat, 1.1% phosphorus, and 3% fiber was used. Plants were harvested every 4 weeks and new seedlings were transplanted. In the first 4-week period, red orache, minutina, and Swiss chard were cultivated, while in the second 4-week period, okahijiki, Swiss chard, and kale were cultivated. Natural daylight and supplemental lighting (high-pressure sodium (HPS) lamps, 600-W, P.L. Light Systems Inc., Beamsville, ON, Canada), were used to support a photoperiod 14 h light (6:00 am to 8:00 pm) and 10 h dark (8:00 pm to 6:00 am). A quantum sensor (LI-250A light meter; LI-COR Biosciences, Lincoln, NE, USA) was used to measure the supplemental photosynthetic photon flux (PPF) in the greenhouse, and the photosynthetically active radiation (PAR) averaged 426 μmol m^−2^ s^−1^. Room temperatures were set at 25 and 20 °C for day and night, respectively, with an hour transition between two temperature regimes.

Probiotics (EZ-Bio; Zeigler Brothers, Gardners, PA, USA) and nitrifying bacteria (Stability; Seachem^®^, Madison, GA, USA) were used to manage water quality and the microbial community within each system. As soon as shrimp were moved into aquaculture tanks, EZ-bio (*Bacillus* spp.) and Stability were inoculated at 10 ppm using recommended doses into each of the 12 systems every day during the acclimation week. Three weeks after transplanting (3rd week of the experiment), EZ-bio and Stability were added again at 10 ppm into each of the 12 systems every day, every other day in the 4th week of the experiment, twice per week in the 5th week of the experiment, and once per week beginning in the 6th week of the experiment, continuing until a week prior to the end of the experiment. pH was adjusted with 10% H_2_SO_4_ or 1M KOH to maintain desired levels (6.5 and 7.5). Water flowed through the 250-micron filter bag to collect solids from the shrimp culture; therefore, plant roots were protected from clogging by biofloc.

The experiment was conducted for 8 weeks (17 April 2021 to 5 June 2021). No water was discharged or exchanged, only evaporative losses were replenished using RO water.

### 2.4. Measurement of Water Quality

During the experiment, dissolved oxygen (DO) and temperature (YSI 55 digital oxygen meter with an integrated thermometer; Xylem Inc., Yellow Springs, OH, USA), and pH (H9813-6; Hanna Instruments, Woonsocket, RI, USA) were measured twice per day at 7 a.m. and 5 p.m. Salinity (Vital Sine™ Salinity Refractometer, Pentair Aquatic Ecosystems, Apopka, FL, USA) was measured once per day at 7 a.m. and adjusted with RO water or sea salt after the last feeding in the afternoon if salinity was not at 15 ppt. Water samples were collected from the aquaculture tank twice a week before feeding, and were analyzed immediately to determine the concentrations of total ammonia nitrogen (TAN), nitrite (NO_2_^−^), nitrate (NO_3_^−^), and phosphate (PO_4_^3−^) using HACH reaction kits (HACH, Loveland, CO, USA). Alkalinity and total suspended solids (TSS) were measured once per week using HACH reaction kits and US EPA method 1684, respectively.

### 2.5. Growth Measurements

#### 2.5.1. Shrimp

Shrimp growth parameters such as initial weight, final weight, the number of shrimp, and the total feed input were collected at the beginning and end of the experiment to determine survival rate, weight gain (WG), and specific growth rate (SGR) using the following formulae:
Survival rate (%) = (Final number of shrimp/Initial number of shrimp) × 100
(1)

Weight gain (%) = (Final biomass (g) − Initial biomass (g))/Initial biomass × 100
(2)

and,
Specific growth rate (%) = [Ln (Final biomass (g)) − Ln (Initial biomass (g))]/day × 100
(3)


#### 2.5.2. Plants

Plant growth parameters such as plant height (cm), leaf length (cm), and SPAD value (which indicates the content of chlorophyll per unit leaf area; SPAD-502 Chlorophyll meter; Minolta Camera Co., Ltd., Osaka, Japan) were measured every two weeks. Leaves that are fully expanded were chosen, we picked 3 leaves per plant and averaged the three data points for each plant. Fv/Fm value is an indication of the maximum efficiency of Photosystem II, and it can be viewed as an indicator of plant stress (fluorescence variable is represented by Fv, and fluorescence maximum by Fm). It was measured before every plant harvest using a Chlorophyll Fluorimeter (Handy EPA^+^; Hansatech Instruments Ltd., King’s Lynn, UK).

At harvest, plant samples were divided into two parts (shoots and roots) and weighed individually for fresh weight. Plant samples were dried in an oven at 70 °C for 72 h and weighed. Initial and final fresh weights were used to calculate relative growth rate (RGR). The water content (WC) in plants was calculated as the difference between fresh weights and dry weights. In addition, dried plant samples were ground and filtered through a 25-mesh sieve and kept in plastic vials for antioxidant and nutrient analysis. Plant tissue analysis was done by the Brookside Laboratory (New Bremen, OH, USA). Extraction of phenolic compounds was performed using 80% (*v*/*v*) methanol and 2% (*v*/*v*) formic acid as extraction solvents. Plant samples (0.2 g) were mixed with 1.5 mL of prepared extraction solution under vortex for 10 min. The mixture was kept on ice for 30 min then stirred for 5 min before centrifugation at 11,000× *g* for 10 min. The supernatant was collected, and the precipitate was mixed with 1 mL of 2% formic acid for the second extraction. The mixture was heated and kept at 60 °C for 15 min then vortexed for 10 min. After centrifugation at 11,000× *g* for 10 min, the supernatant was combined with the supernatant from the first centrifugation and stored for further analysis. The total phenolic content (TPC) was quantified by the Folin–Ciocalteu micro-method [38] following the procedure of [39,40]. In brief, phenolic extract (35 μL) was mixed with 150 μL diluted Folin–Ciocalteu’s reagent and 115 μL of 7.5% (*w*/*v*) Na_2_CO_3_. Then, the mixture was kept at 45 °C for 30 min followed by additional 1 h incubation at room temperature. The absorbance of sample was measured at 765 nm wavelength and result was expressed as milligram of gallic acid equivalent (GAE) per gram of dry matter (gdm). Antioxidant capacity was measured via two assays, 2,2-diphenyl-1-picryl-hydrazyl-hydrate (DPPH) free radical-scavenging activity [41], and 2,2-azino-bis (3-ethylbenzothiazoline-6-sulfonic acid) (ABTS) scavenging activity [42]. Briefly, 15 μL of extract was added into 285 μL of DPPH solution and mixed at 400 rpm. The absorbance at 515 nm was recorded after 2 h of incubation in dark. For the ABTS assay, 10 μL of sample was mixed with 294 μL of diluted ABTS solution then the mixture was incubated at 30 °C for 10 min. The absorbance was measured at 734 nm. Trolox was used as the standard for both assays. Formulae used to calculate plant growth indices and nutrient use efficiency are shown below:Relative growth rate (%) = [Ln (Final biomass (g)) − Ln (Initial biomass (g))]/day × 100
(4)
Water content (%) = (Final fresh weight (g) − Final dry weight (g))/Final fresh weight × 100
(5)

and,
Nutrient use efficiency = (g nutrient absorbed)/(g nutrient supplied) × 100
(6)

### 2.6. Statistical Analysis

Shrimp and plant growth parameters, nutrient and antioxidant concentrations in plants, solid wastes, and water quality parameters were analyzed using JMP Pro 16.0 (SAS Institute Inc., Cary, NC, USA) and treatment means compared by two-way analysis of variance (ANOVA). If ANOVA indicated significant treatment effects, differences between means were determined by Tukey’s honestly significant difference test (HSD) at *p* ≤ 0.05 [43].

## 3. Results

### 3.1. Shrimp

There were no significant (*p* > 0.05) differences in final weight, weight gain (WG), specific growth rate (SGR) or survival of shrimp among all treatments (Table 1). Based on the two-way ANOVA, survival was not affected by the interaction between pH and additional C or additional C, but significantly (*p* < 0.05) affected by pH. Other growth parameters (final weight, WG, and SGR) were not affected by the interaction between factors, pH, or additional C. Higher final weight, weight gain, and specific growth rate were in the 6.5 + C treatment, followed by 7.5 No C, 6.5 No C, and 7.5 + C treatments. Survival was higher in 6.5 No C treatment (54.7%), followed by 42.8, 27.4, and 19.0% in the 6.5 + C, 7.5 No C, and 7.5 + C treatment, respectively.

### 3.2. Plants

#### 3.2.1. Growth and Yield of Plant

Overall, in both harvests, the interaction between the two factors affected most plant species on fresh weight, dry weight, and yield significantly (*p* < 0.05), except for okahijiki. Whereas pH and the additional C significantly (*p* < 0.05) affected plants’ fresh weight, dry weight, and yield, except for minutina (Table 2 and Table 3). Generally, growth parameters of all plants (Table 4 and Table 5) were significantly affected by the interaction of the two factors, pH, and additional C. All plant species grown in 7.5 No C treatment had significantly lower (*p* < 0.05) growth than the other treatments (Figure 2 and Figure 3).

Both red orache and okahijiki were sensitive to higher pH and growth was enhanced by additional C. Plants grown in 6.5 + C treatment had significantly higher (*p* < 0.05) growth (fresh weight, dry weight, plant height, RGR, and yield), SPAD value, and Fv/Fm. Red orache grown in the 7.5 + C treatment had similar results to 6.5 No C treatments and significant (*p* > 0.05) difference with those grown in 7.5 No C treatment. Although additional C enhanced the growth performance of okahijiki, plants in the 7.5 + C treatment still had significantly lower (*p* < 0.05) performance than both 6.5 treatments. Minutina, Swiss chard, and kale, however, were unaffected by higher pH. Some of them even grew effectively at higher pH, but additional C was required.

The addition of C increased root growth (Figure 2 and Figure 3), which could lead to greater nutrient uptake. Chlorotic and necrotic leaves were more common among plants harvested from the 7.5 No C treatment, which indicated there were not enough nutrients for them. All treatments displayed similar leaves and roots of the minutina (Figure 2), but it was obvious that plants in both pH 7.5 treatments had begun reproductive growth. This phenomenon might be triggered by abiotic stress.

#### 3.2.2. Mineral Nutrients, Total Phenolics Content, and Antioxidant Capacity

The interaction of the two factors exerted effects on red orache and minutina’s N and P content, and Swiss chard’s N, P, NUE, and PUE results, and the content of P and NUE in kale, but no effects on okahijiki. In general, the content of N and P in plant tissue and NUE, and PUE were significantly (*p* < 0.05) impacted by pH and additional C (Table 6 and Table 7). The concentration of N was significantly higher (*p* < 0.05) in red orache, minutina, and Swiss chard (Table 6) cultivated in 7.5 No C treatment in the first harvest (Table 6). However, there was no significant (*p* > 0.05) differences found in NUE among treatments. In the second harvest (Table 7), although the N content in okahijiki was higher in pH 7.5 treatments, it was not significantly (*p* > 0.05) different among treatments. Kale and Swiss chard had significantly (*p* < 0.05) higher N in tissues in pH 6.5 treatments; however, with additional C, plants grown at pH 7.5 had similar results to pH 6.5 treatments. Further, plants harvested from the second batch had better NUE in pH 6.5 treatments. Again, the additional C improved the performance of plant NUE in pH 7.5 treatment.

Plants grown at pH 6.5 had higher P concentration and PUE, yet, some plants, such as minutina and kale, cultured at pH 7.5 with additional C exhibited no difference in P concentration and PUE with those plants grown at pH 6.5, which was similar to the results found in plant growth.

Figure 4 showed the total phenolics concentration (TPC) and antioxidant capacity in plants grown in two pH levels with or without additional C in the culture environment. The interaction of the two factors only affected TPC in red orache and Swiss chard, and TPC in plants was significantly (*p* < 0.05) affected by pH and additional C in general (Table 8). The antioxidant capacity of plants from the first harvest was affected by the interaction between the two factors, pH, and additional C in general (Table 8). In terms of second harvest, there was no interaction effect of the two factors on all three species (Table 8). Only okahijiki and Swiss chard were affected by pH and additional C, respectively (Table 8). TPC and antioxidant capacity showed similar trends in most plants from both harvests. Red orache and minutina had higher TPC and antioxidant capacity in both pH 6.5 treatments and 7.5 + C treatments than in 7.5 No C treatment. Okahijiki and Swiss chard (first harvest) had significantly higher (*p* < 0.05) TPC and antioxidant capacity in 6.5 + C treatment than other treatments, while Swiss chard from the second harvest had higher TPC and antioxidant capacity in + C treatments than No C treatments. Different from other plant species, the highest TPC and antioxidant capacity in kale were found in 7.5 No C treatment, followed by 7.5 + C treatment, 6.5 No C treatment, and then 6.5 + C treatment.

### 3.3. Water Quality

pH was maintained at an average of 6.5 or 7.5 in systems throughout the experiment. Dissolved oxygen (DO) and temperature were maintained between 6.4 to 7.3 mg/L and 26.7 to 28.3 °C, respectively. Salinity was controlled at an average of 15 ppt in all treatments. Daily loss of water through evaporation and transpiration was roughly 3.7 to 4.0 % of the total volume of water in each system. The concentration of total suspended solids (TSS) in most treatments remained low throughout the experiment (Figure 5A). However, the concentration of TSS in the 6.5 + C treatment started to increase beginning the third week of the experiment, and the TSS in 7.5 + C and 7.5 No C treatments increased in the last week of the experiment. Overall, there were no significant (*p* > 0.05) differences found in TSS among treatments during the entire experiment. Alkalinity in both pH 7.5 treatment was significantly higher (*p* < 0.05) than pH 6.5 treatments (Figure 5B). The 6.5 No C treatment had the lowest alkalinity throughout the experiment; however, the alkalinity in 6.5 + C treatment steady increased with no difference with 7.5 No C treatment after 5 weeks.

The concentration on TAN at the beginning of the experiment was significantly higher (*p* > 0.05) in treatments without additional C (Figure 5C). The concentration of TAN in 7.5 No C, 7.5 + C, 6.5 + C, and 6.5 No C treatments decreased to safe levels on day 7, 10, 17, and 21, respectively. However, TAN concentration in both pH 6.5 treatments increased after the first harvest (day 28). After TAN decreased, NO^2−^ started to increase (Figure 5D). The concentration of NO^2−^ in 7.5 No C increased rapidly with the highest concentration (19 mg/L) on day 21, but did not decline to a safe level (<1 mg/L) until day 52. In contrast, NO^2−^ in other treatments declined to safe levels between day 21 and 28. However, similar to TAN, the concentration of NO^2−^ in all treatments increased again and was not back to the safe range until the end of the experiment. The change in NO^3−^ concentrations in all treatments was similar to the NO^2−^ trend (Figure 5E). The concentration of PO_4_^3−^ steadily increased in all treatments (Figure 5F). 7.5 + C and 6.5 + C treatments had significantly higher (*p* < 0.05) PO_4_^3−^ than 7.5 No C and 6.5 No C treatments throughout the experiment.

## 4. Discussion

### 4.1. Shrimp Growth

The pH range of 7.0 to 9.0 is acceptable for shrimp; however, the optimal range suggested for shrimp cultured in recirculating systems is 7.2 to 7.8 [44]. Beyond the acceptable range such as very acidic water, where pH < 6.5, or very basic water where pH > 10.0, physiological systems, such as ion regulation, respiration, enzyme activities, and immunity are influenced, subsequently weakening the antioxidant ability, causing DNA damage, increasing susceptibility to diseases, and eventually retarding growth and reducing survival [19,44,45,46,47,48,49]. Due to the impaired ionic regulation in the low pH environment, diffusion gradients for NH_3_ should be increased increasing ammonia excretion and alleviating toxicity in vivo [50]. However, in the present study, the TAN concentrations in the lower pH treatments were over the safe level (2 mg/L at 15 ppt; [51]) most of the time in the experiment. Even though the toxicity of TAN is low at pH 6.5, the high level of TAN in the water lowered the excretion of ammonia and accumulated in the body and contributed to toxicity to shrimp. This might be one of the factors contributing to the low survival in pH 6.5 treatments in the present study. On the other hand, although the TAN concentration in pH 7.5 treatments remained low, the concentration of NO_2_^−^, another toxic nitrogenous compound for aquatic animals, was over the safe level (6 mg/L at 15 ppt; [52]) during the experiment. Under high NO_2_^−^ concentration, oxygen (O_2_) transfer is reduced as NO_2_^−^ competes the oxygen active site of copper with O_2_ and forms the meta-hemocyanin, which is nonfunctional [49,53]. This phenomenon might be one of the potential reasons of the high mortality in pH 7.5 in the present study. With nonfunctional meta-hemocyanin, shrimp was not able to obtain enough oxygen, and this might explain why the survival was affected by pH and showed a lower level in pH 7.5 treatments than in pH 6.5 treatments.

In addition, the ionic composition in water is another important factor to optimize the shrimp culture. Boyd [54] and Davis et al. [55], suggested that maintaining K, Ca, and Mg levels similar to the levels in seawater diluted to the same salinity is desired when culturing shrimp with groundwater, surface water, or spring water at salinity lower than seawater. In the present study, RO water, which has nearly no minerals or ions, was used. Although the sea salt applied in the present study has the required amount of Ca and Mg needed as recommended, in aquaponics, shrimp, plants, and microorganisms all require minerals for their growth and metabolism. Shrimp, especially, lose minerals through molting and need minerals from the environment to calcify their cuticles [56,57]. During the experiment, Ca and Mg was constantly supplemented via feed input; however, the amount of mineral might not be enough for all three organisms and impacts the calcification of the cuticle and delays recovery of shrimp from molting [57,58], leading to high mortality in all treatments. More research is needed to investigate the strategy of mineral management in aquaponics, which uses RO water or water sources that are ionic composition imbalanced.

### 4.2. Plants

For plants, pH plays a significant role in nutrient availability, with lower pH preferred for most species and most ions [4]. This could explain the better plant growth in pH 6.5 treatments. Nevertheless, amending the growing environment with molasses to improve soil fertility and increase crop yield was recognized in agriculture for many years due to the rich mineral elements and organic matter [59]. However, carbon, the primary constituent of molasses, is the key element that contributes to a better crop yield as carbohydrates are the main energy substrates for plant respiration; in addition, their movement in phloem is the main force of nutrient transportation [60]. Although the primary source of C for plants is assimilated from atmospheric sources by leaves, the addition of C, which was provided into culture water, uptake by root can improve nutrient uptake, metabolism, and growth [22,23,59,60,61]. The better growth, yield, and higher P concentration in + C treatments may be attributed to the addition of molasses, which provide a high concentration of C to improve nutrient assimilation. The result of the present suggest that under saline conditions and/or high environmental pH, providing external C could improve plant’s growth and yield.

Nutrient content analysis revealed that N concentration was significantly higher in plants that were grown in the pH 7.5 No C treatments; treatments that displayed lower growth. This phenomenon might be the dilution effect [62]. Plants provided with better environmental conditions or higher nutrients produce greater yields; however, chemical analysis showed that the average amount of the element in some or all plant tissue is lower than in deficient control plants [63,64,65]. This is because as plants thrive in suitable environments, most nutrients, particularly N, are used for growth, causing increased production of dry matter, whereas nutrients can accumulate in suboptimal conditions where growth is restricted [62].

The SPAD value (an index of chlorophyll content per unit leaf area) and the chlorophyll fluorescence parameter Fv/Fm (maximum photochemical efficiency of PSII) are two important indicators for plant health and stress, respectively [66,67]. For most plants, Fv/Fm values are close to and/or higher than 0.83 representing growing without stress [66,67]. The result of SPAD value and Fv/Fm in the present study showed a similar trend with plant growth. Plants grown in 7.5 No C treatment showed significantly lower (*p* < 0.05) values of SPAD and Fv/Fm; lower than 0.83. Based on this result, we suggest that plants were stressed under a high environmental pH and growth was affected, while providing additional C can be a promising approach to alleviate stress and improve growth. Further research is needed to discover how the C improves plant growth under abiotic stress. Compared with other halophytes, minutina might be a more promising candidate for development in marine aquaponics, as its Fv/Fm value in 7.5 No C treatment was not significantly different from other treatments, which indicates that it has a higher tolerance than the other two halophytes.

In addition to the growth response and nutrient content of plants, the amount of antioxidants in plants is another factor worth evaluating when it comes to the production of food. Natural antioxidants are primarily found in plants, and most of them are phenolics, which are secondary metabolites that have a variety of health-promoting effects in humans [68,69]. The value of these compounds lies in their primary antioxidant activity, including their role as free radical acceptors and chain breakers [68]. Plant species, analysis method, and environmental factors can affect phenolic compound concentration [70,71,72,73,74,75,76]. According to the results of the present study, TPC in most plant species was higher in the lower pH treatments, which was similar to the results reported by Alexopoulos et al. [70] and Radić et al. [71]. TPC could be increased if plants are grown with some abiotic stress; however, the lowest TPC was found in most plants grown in the 7.5 No C treatment. The lower nutrient availability might be the factor to this result [70].

### 4.3. Water Quality

The environmental pH that the two distinct types of bacteria, nitrifying bacteria and heterotrophic bacteria, prefer is 7.0 to 8.0 [77,78,79,80,81]. Growth is limited and their ability to convert or remove toxic nitrogen compounds is reduced if the pH is not within that range [82]. It took 12, 20, and 24 days for nitrifying bacteria to reduce TAN from 5 to 0 mg/L at pH 8.5, 7.5, and 6.5, respectively, and no nitrification was found at pH 5.5 [82]. Wongkiew et al. [83] reported that higher TAN and lower nitrite and nitrate concentrations were found in lower pH treatments, which is similar to the present study. Moreover, comparing the results between 7.5 + C and 7.5 No C treatments, which were at the proper pH range for nitrifying bacteria, the efficiency of nitrification in 7.5 No C treatment was relatively higher due to lower C/N ratio and less competition with heterotrophic bacteria. Consequently, the TAN was in a safe range earlier than the other treatments. However, 7.5 No C treatment has the highest level of NO_2_^−^, which might be because the growth rate of nitrite-oxidizing bacteria was slower than that of ammonia-oxidizing bacteria, which resulted in the rate of oxidizing NO_2_^−^ being lower than the rate of NO_2_^−^ generation. Concentrations of TAN and NO_2_^−^ in both pH 6.5 treatments decreased below the safe range around day 17 to 28 (Figure 5C,D), largely attributed to plants’ ability to absorb nitrogenous compounds when plants were near the harvest size.

After the first harvest (day 28), the concentration of TAN and NO_2_^−^ increased in all treatments in the present study (Figure 5C,D), which is similar to results from other studies [21,84]. Chu and Brown [9] reported that the concentration of nitrogenous compounds increased after harvests, but the concentration of TAN did not increase to a hazardous level at the late stage, and the concentration of NO_2_^−^ remained within safe ranges throughout the experiment. In the present study, the potential reasons for TAN and NO_2_^−^ increasing beyond the safe range after harvest might be the frequency of inoculating probiotics, the C/N ratio, animal to plant ratio, the water source, or the ionic composition in the water. Chu and Brown [9,21,31] inoculated probiotics on a regular basis and maintained the C/N ratio at 12 and 15, which provided a better environment for heterotrophic bacteria to assimilate nitrogenous compounds. The lower C/N ratio and discontinuous inoculation in the present study could be one of the causes of the high concentration of TAN and NO_2_^−^. Further, the stocking density ratio of shrimp to plant in this study was 5.5 to 1, which is higher than the suggested ratio in Chu and Brown [9], which could be another cause for the hazard level of toxic nitrogen compounds.

Additionally, water source and the ionic composition are potential factors related to the results of water quality in the present study. RO water, which has nearly no minerals or ions, was used in the experiment. However, growth of microorganisms requires Mg, P, Fe, Ca, S, and K, especially the first three elements [85]. Without sufficient minerals, microbes are not able to sustain growth and metabolism, not to mention the major function of converting toxic nitrogen compounds. Although minerals such as Na, Cl, Ca, Mg, S, and K were dissolved in the water during salinity adjustment or feed input, their concentrations may not be sufficient to support the demand from the three organisms [54,85,86]. We suggested that systems using RO water as the major water source, monitoring and managing ionic composition is necessary. More research is needed to determine the effect of ion management and the water source on the practice of aquaponics.

## 5. Conclusions

The current study found no significant effects of pH or additional C on shrimp performance in final weight, weight gain (WG), and specific growth rate (SGR). Yet, pH affected survival and showed lower levels in pH 7.5 treatments. In contrast, plants grew better in the lower pH treatments, while additional C supplements improved the performance of plants grown in the higher pH treatments and had similar results to the lower pH treatments. The addition of C led to improved growth and yields of most plants. Hence, adding C can be a promising approach in marine aquaponics to enhance the resistance to the abiotic stress of plants and improve their growth. Applying additional C is suggested as a solution of pH conundrum for the operation of marine aquaponic food production system when the pH is high. Nevertheless, further research is needed to understand ion management in aquaponics for better management of microbial colonies, water quality, and better cultivation of shrimp and plants in the system.

## Figures and Tables

**Figure 1 foods-12-00069-f001:**
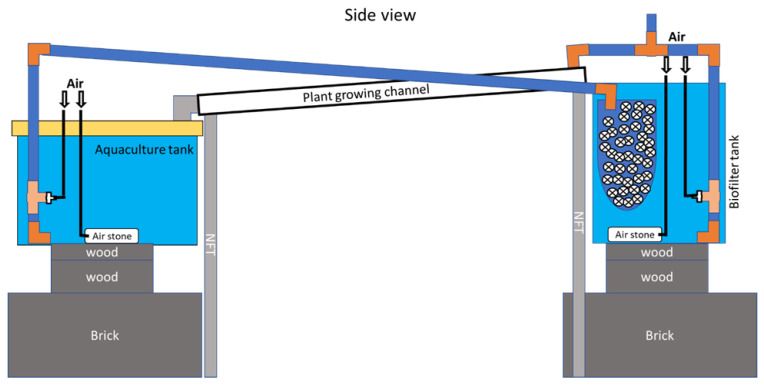
Schematic diagram of NFT aquaponic systems.

**Figure 2 foods-12-00069-f002:**
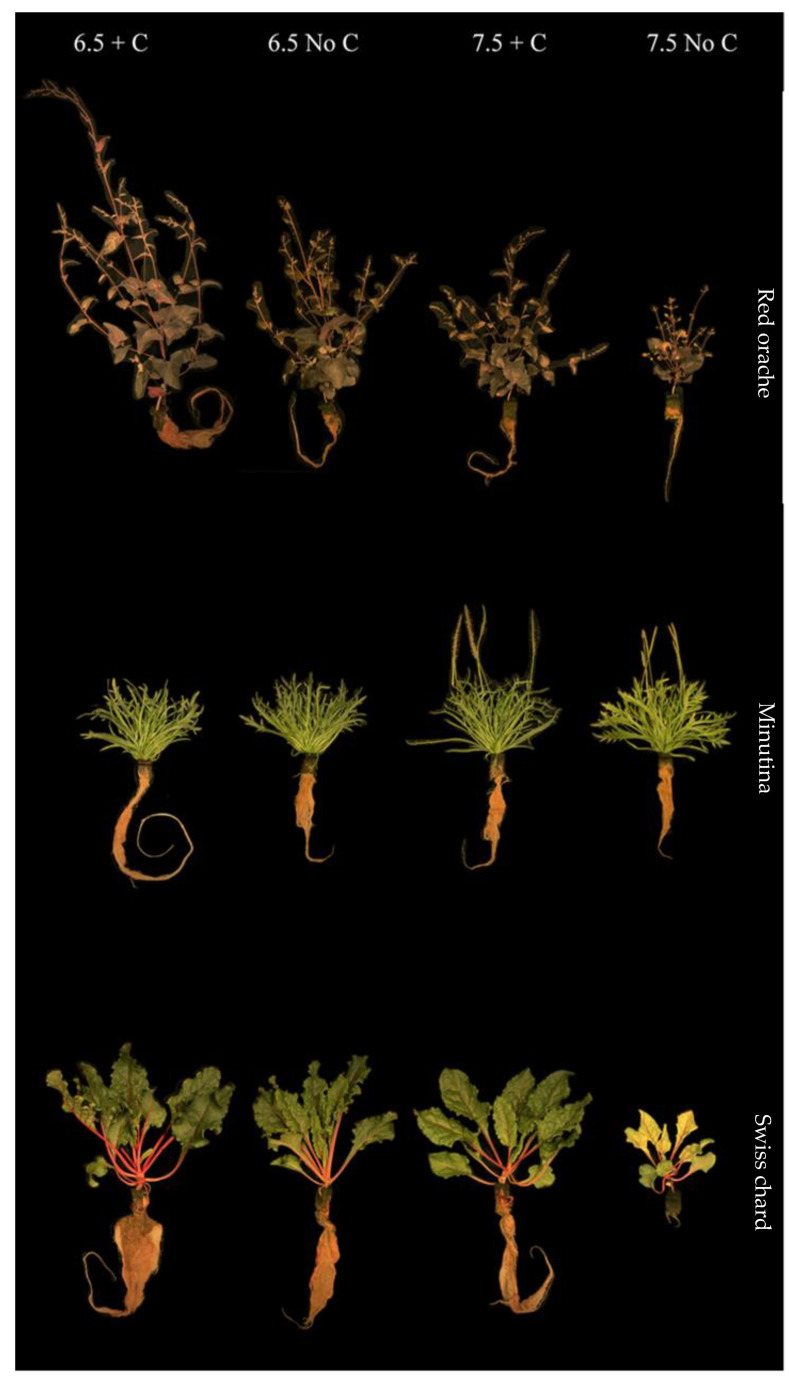
Plants harvested from the first 4 weeks.

**Figure 3 foods-12-00069-f003:**
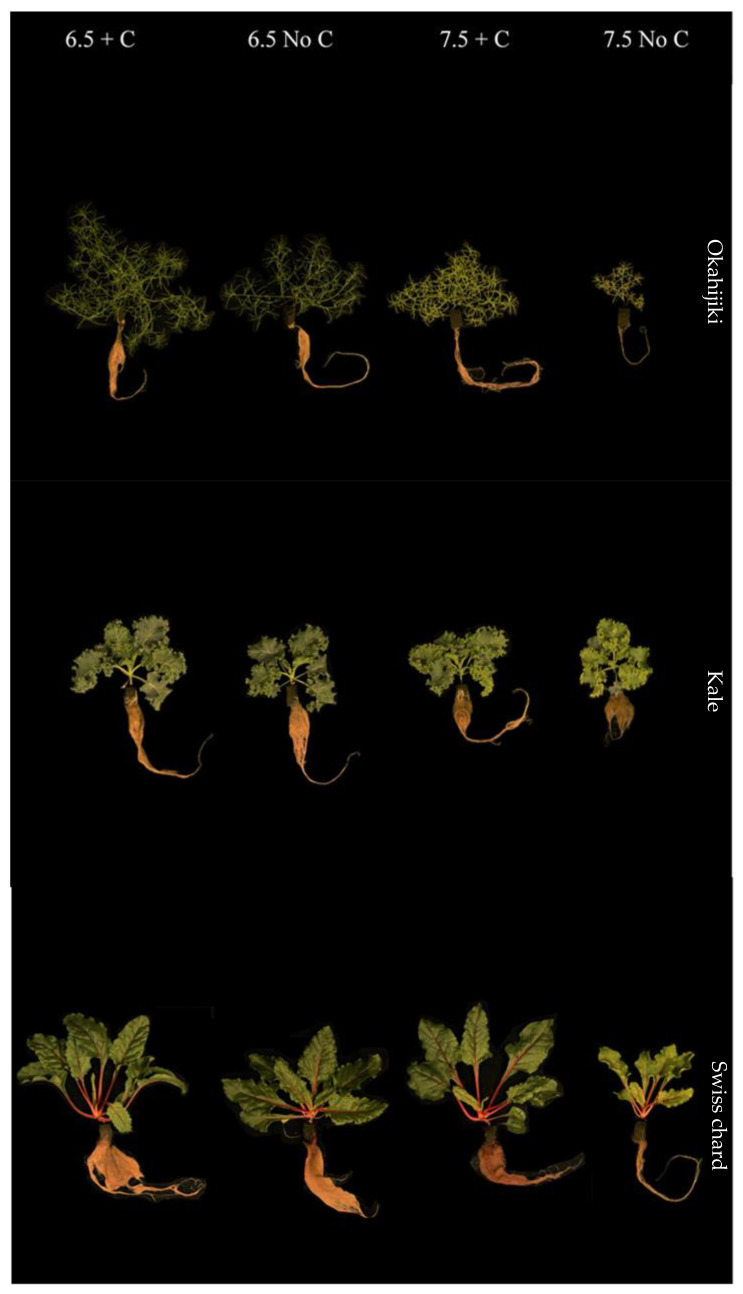
Plants harvested from the second 4 weeks.

**Figure 4 foods-12-00069-f004:**
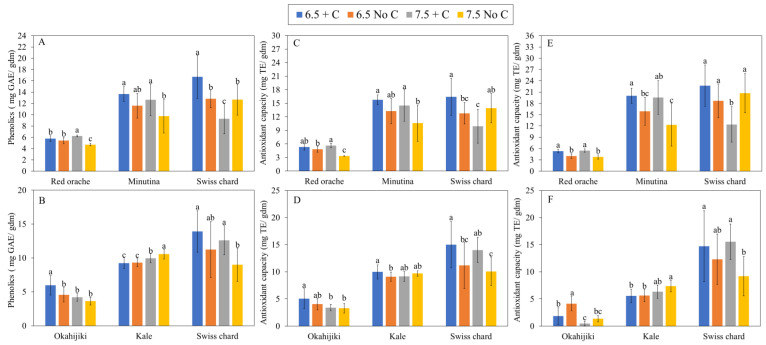
Total phenolic content of plants from first and second harvests are (**A**) and (**B**), respectively. Antioxidant capacity of plants from first and second harvests measured via ABTS and DPPH are (**C**) and (**D**), and (**E**) and (**F**), respectively. Different letters above the bars within plant species indicate significant difference based on Tukey’s honestly significant difference test (α = 0.05).

**Figure 5 foods-12-00069-f005:**
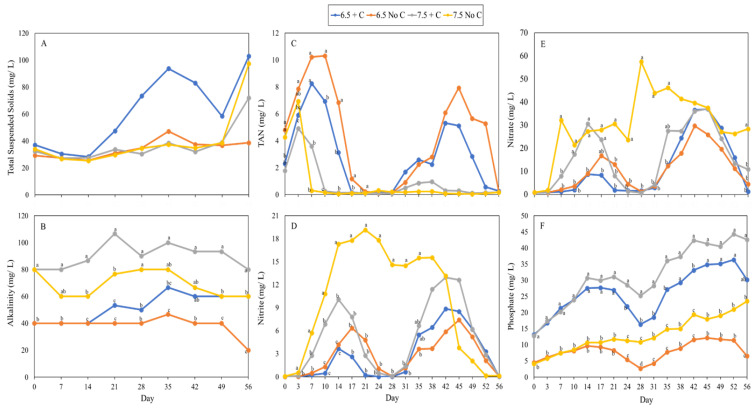
The change of total suspended solids (**A**), alkalinity (**B**), TAN (**C**), nitrite (**D**), nitrate (**E**), and phosphate (**F**) concentrations in marine aquaponics under two pH levels with or without additional C for 8 weeks. Lowercase alphabet letters represent significant differences, followed by one way ANOVA and Tukey’s HSD test (α = 0.05).

**Table 1 foods-12-00069-t001:** Response of shrimp grown in marine aquaponics at two pH levels with or without additional carbon for 8 weeks.

Parameter	Treatment							ANOVA	
	pH	6.5		7.5		*P*	pH	Additional C	pH × Additional C
	Additional C	+C	No C	+C	No C				
Initial number of shrimp per tank		28 ± 0	28 ± 0	28 ± 0	28 ± 0	ns	ns	ns	ns
Initial Weight (g) per shrimp		2.25 ± 0.55	2.25 ± 0.58	2.25 ± 0.55	2.24 ± 0.58	ns	ns	ns	ns
Final number of shrimp per tank		12 ± 3	15 ± 5	5 ± 7	8 ± 4	ns	*	ns	ns
Final Weight (g) per shrimp		8.29 ± 0.64	7.66 ± 1.17	7.35 ± 2.01	8.03 ± 0.38	ns	ns	ns	ns
WG (%)		268.5 ± 28.4	241.1 ± 51.7	226.2 ± 88.2	258.3 ± 17.1	ns	ns	ns	ns
SGR (%)		2.33 ± 0.14	2.18 ± 0.26	2.07 ± 0.45	2.28 ± 0.09	ns	ns	ns	ns
Survival (%)		42.8 ± 9.5	54.7 ± 18.0	19.0 ± 23.8	27.4 ± 12.5	ns	*	ns	ns

Each value represents means ± SD. ns and * mean no significant or significant at *p* ≤ 0.05, respectively.

**Table 2 foods-12-00069-t002:** Fresh and dry weight of red orache, minutina, and Swiss chard in aquaponics under two pH levels with or without additional C in the first harvest.

Plant Species	Treatment		Fresh Weight (g/Plant)			Dry Weight (g/Plant)			Yield (kg/m^2^)
	pH	Additional C	Total	Shoot	Root	Total	Shoot	Root	
Red orache	6.5	+C	42.7 ± 10.2 a	23.3 ± 3.2 a	18.1 ± 7.4 a	4.97 ± 0.85 a	3.22 ± 0.49 a	1.75 ± 0.50 a	0.53 ± 0.02 a
		No C	24.8 ± 6.6 b	19.3 ± 3.8 b	4.0 ± 1.3 bc	3.29 ± 0.82 b	2.64 ± 0.56 b	0.55 ± 0.18 bc	0.46 ± 0.10 a
	7.5	+C	24.2 ± 7.1 b	18.1 ± 3.9 b	6.4 ± 3.7 b	3.38 ± 1.01 b	2.53 ± 0.61 b	0.85 ± 0.44 b	0.41 ± 0.09 a
		No C	12.0 ± 3.2 c	9.0 ± 2.2 c	2.6 ± 0.6 c	1.65 ± 0.46 c	1.28 ± 0.39 c	0.37 ± 0.07 c	0.21 ± 0.06 b
	*p*		***	***	***	***	***	***	**
ANOVA									
pH			***	***	***	***	***	***	**
Additional C			***	***	***	***	***	***	*
pH × Additional C			ns	**	***	ns	**	***	ns
Minutina	6.5	+C	45.3 ± 12.7 ab	37.0 ± 9.8 ab	8.3 ± 5.3	3.83 ± 1.10 ab	3.17 ± 0.92 ab	0.66 ± 0.32	0.85 ± 0.16
		No C	55.5 ± 10.9 a	46.2 ± 9.9 a	9.3 ± 4.7	4.56 ± 0.94 a	3.85 ± 0.92 a	0.67 ± 0.20	0.99 ± 0.09
	7.5	+C	53.4 ± 19.1 ab	43.3 ± 17.4 ab	10.1 ± 5.3	4.19 ± 1.39 ab	3.44 ± 1.27 ab	0.76 ± 0.29	0.99 ± 0.17
		No C	41.2 ± 17.1 b	32.1 ± 13.6 b	9.1 ± 4.6	3.30 ± 1.41 b	2.63 ± 1.18 b	0.67 ± 0.24	0.73 ± 0.30
	*p*		*	*	ns	*	*	ns	ns
ANOVA									
pH			ns	ns	ns	ns	ns	ns	ns
Additional C			ns	ns	ns	ns	ns	ns	ns
pH × Additional C			**	**	ns	**	**	ns	ns
Swiss chard	6.5	+C	50.5 ± 9.6 a	36.5 ± 10.7 a	15.3 ± 5.3 b	5.25 ± 0.81 a	3.88 ± 1.12 a	1.38 ± 0.35 b	0.83 ± 0.11 ab
		No C	59.6 ± 20.2 a	40.3 ± 12.4 a	17.8 ± 6.1 ab	5.22 ± 1.46 a	4.05 ± 1.22 a	1.35 ± 0.36 b	0.92 ± 0.08 ab
	7.5	+C	66.3 ± 26.5 a	43.617.0± a	23.5 ± 10.2 a	6.13 ± 2.10 a	4.35 ± 1.59 a	1.78 ± 0.55 a	0.99 ± 0.27 a
		No C	26.3 ± 11.1 b	22.4 ± 10.0 b	3.9 ± 1.3 c	2.91 ± 0.94 b	2.42 ± 0.85 b	0.49 ± 0.11 c	0.51 ± 0.16 b
	*p*		***	***	***	***	***	***	*
ANOVA									
pH			*	ns	ns	*	*	*	ns
Additional C			***	**	***	***	**	***	ns
pH × Additional C			***	***	***	***	***	***	*

Each value represents mean (*n* = 18). Values in the same columns of each plant species with different lowercase alphabet letter are significantly different (*p* < 0.05). ns, *, **, *** mean no significant or significant at *p* ≤ 0.05, respectively.

**Table 3 foods-12-00069-t003:** Fresh and dry weight of okahijiki, kale, and Swiss chard in aquaponics under two pH levels with or without additional C in the second harvest.

Plant Species	Treatment		Fresh Weight (g/Plant)			Dry Weight (g/Plant)			Yield (kg/m^2^)	
	pH	Additional C	Total	Shoot	Root	Total	Shoot	Root		
Okahijiki	6.5	+C	30.3 ± 8.6 a	26.3 ± 8.1 a	4.0 ± 1.5 a	2.91 ± 0.82 a	2.44 ± 0.78 a	0.49 ± 0.09 a	0.64 ± 0.14 a	
		No C	22.1 ± 9.7 b	17.9 ± 9.2 b	4.2 ± 2.1 a	1.82 ± 0.78 b	1.47 ± 0.72 b	0.33 ± 0.07 b	0.41 ± 0.17 ab	
	7.5	+C	10.6 ± 3.7 c	8.2 ± 3.0 c	2.7 ± 1.2 b	1.07 ± 0.35 c	0.69 ± 0.22 c	0.32 ± 0.07 b	0.22 ± 0.07 bc	
		No C	2.9 ± 1.3 d	1.9 ± 1.1 d	1.1 ± 0.3 c	0.43 ± 0.11 d	0.20 ± 0.08 c	0.23 ± 0.04 c	0.05 ± 0.02 c	
	*p*		***	***	***	***	***	***	**	
ANOVA										
pH			***	***	***	***	***	***	***	
Additional C			***	***	ns	***	***	***	*	
pH × Additional C			Ns	ns	*	ns	ns	*	ns	
Kale	6.5	+C	27.0 ± 5.0 ab	21.1 ± 3.7 a	5.9 ± 1.7 ab	2.96 ± 0.46 a	2.42 ± 0.37 a	0.52 ± 0.10 a	0.48 ± 0.06 a	
		No C	29.2 ± 5.3 a	22.3 ± 3.9 a	6.8 ± 2.0 a	2.96 ± 0.46 a	2.42 ± 0.37 a	0.52 ± 0.10 a	0.51 ± 0.06 a	
	7.5	+C	25.2 ± 3.3 b	19.9 ± 2.7 a	5.2 ± 1.1 b	2.72 ± 0.35 a	2.21 ± 0.28 a	0.51 ± 0.09 a	0.46 ± 0.02 a	
		No C	16.1 ± 3.0 c	13.0 ± 2.5 b	3.0 ± 0.5 c	1.76 ± 0.32 b	1.35 ± 0.27 b	0.36 ± 0.03 b	0.30 ± 0.02 b	
	*p*		***	***	***	***	***	***	**	
ANOVA										
pH			***	***	***	***	***	***	**	
Additional C			***	***	ns	***	***	***	*	
pH × Additional C			***	***	***	***	***	***	**	
Swiss chard	6.5	+C	50.2 ± 13.4 a	37.0 ± 10.3 a	13.3 ± 3.7 a	4.54 ± 1.22 a	3.53 ± 1.04 a	1.02 ± 0.24 a	0.85 ± 0.09 ab	
		No C	54.4 ± 19.8 a	38.6 ± 13.0 a	13.7 ± 4.8 a	4.78 ± 1.61 a	3.82 ± 1.32 a	1.00 ± 0.31 a	0.93 ± 0.10 a	
	7.5	+C	51.8 ± 20.2 a	38.3 ± 13.8 a	11.2 ± 4.2 a	4.86 ± 1.73 a	3.89 ± 1.43 a	0.97 ± 0.34 a	0.93 ± 0.26 a	
		No C	26.0 ± 9.1 b	21.5 ± 7.9 b	4.5 ± 1.5 a	2.26 ± 0.38 b	1.80 ± 0.31 b	0.50 ± 0.11 b	0.49 ± 0.14 b	
	*p*		***	***	***	***	***	***	*	
ANOVA										
pH			***	**	***	**	**	***	ns	
Additional C			**	**	***	***	**	***	ns	
pH × Additional C			***	**	***	***	***	**	*	

Each value represents mean (*n* = 18). Values in the same columns of each plant species with different lowercase alphabet letter are significantly different (*p* < 0.05). ns, *, **, *** mean no significant or significant at *p* ≤ 0.05, respectively.

**Table 4 foods-12-00069-t004:** Plant growth parameters of red orache, minutina, and Swiss chard in aquaponics under two pH levels with or without additional C in the first harvest.

Plant Species	Treatment		Plant Height (cm)	Leaf Length (cm)	SPAD	Fv/Fm	RGR ^x^ (%)
	pH	Additional C					
Red orache	6.5	+C	49.7 a	-	48.2 a	0.83 a	11.4 a
		No C	38.3 b	-	46.7 ab	0.82 a	11.8 ab
	7.5	+C	37.6 b	-	44.8 b	0.82 a	10.4 b
		No C	16.4 c	-	39.9 c	0.76 b	8.0 c
	*p*		***	-	***	***	***
ANOVA							
pH			***	-	***	***	***
Additional C			***	-	***	***	***
pH × Additional C			***	-	*	***	***
Minutina	6.5	+C	-	-	-	0.83	16.4 a
		No C	-	-	-	0.83	17.1 a
	7.5	+C	-	-	-	0.83	16.9 a
		No C	-	-	-	0.83	12.3 b
	*p*		-	-	-	ns	***
ANOVA							
pH			-	-	-	ns	***
Additional C			-	-	-	ns	***
pH × Additional C			-	-	-	ns	***
Swiss chard	6.5	+C	-	13.9 a	46.1 a	0.83 ab	13.6 a
		No C	-	12.6 a	45.9 a	0.84 a	13.8 a
	7.5	+C	-	14.1 a	42.5 a	0.84 a	13.9 a
		No C	-	9.1 b	33.3 b	0.82 b	11.0 b
	*p*		-	***	***	**	***
ANOVA							
pH			-	***	***	ns	***
Additional C			-	***	**	ns	***
pH × Additional C			-	**	**	***	***

Values in the table are mean (*n* = 18 for plant height, leaf length, SPAD, and RGR; *n* = 9 for Fv/Fm). Values in the same columns of each plant species with different lowercase alphabet letter are significantly different (*p* < 0.05). ns, *, **, *** mean no significant or significant at *p* ≤ 0.05, respectively. ^x^ Relative growth rate. - means no data.

**Table 5 foods-12-00069-t005:** Plant growth parameters of okahijiki, kale, and Swiss chard in aquaponics under two pH levels with or without additional C in the second harvest.

Plant Species	Treatment		Plant Height (cm)	Leaf Length (cm)	SPAD	Fv/Fm	RGR ^x^ (%)
	pH	Additional C					
Okahijiki	6.5	+C	-	-	-	-	15.5 a
		No C	-	-	-	-	13.5 b
	7.5	+C	-	-	-	-	11.5 c
		No C	-	-	-	-	5.8 d
	*p*		-	-	-	-	***
ANOVA							
pH			-	-	-	-	***
Additional C			-	-	-	-	***
pH × Additional C			-	-	-	-	***
Kale	6.5	+C	3.2 a	-	49.5 ab	0.83	10.2 a
		No C	2.8 b	-	50.2 a	0.84	10.4 a
	7.5	+C	3.1 ab	-	45.8 b	0.83	10.0 a
		No C	2.5 c	-	34.7 c	0.83	8.5 b
	*p*		***	-	***	ns	***
ANOVA							
pH			**	-	***	ns	***
Additional C			***	-	***	ns	***
pH × Additional C			ns	-	***	ns	***
Swiss chard	6.5	+C	-	14.0 a	45.5 a	0.85 a	13.5 a
		No C	-	14.1 a	43.7 ab	0.84 a	13.9 a
	7.5	+C	-	13.7 a	41.4 b	0.85 a	13.7 a
		No C	-	10.5 b	34.5 c	0.82 b	11.5 b
	*p*		-	***	***	***	***
ANOVA							
pH			-	***	***	*	***
Additional C			-	**	***	***	**
pH × Additional C			-	***	**	**	***

Values in the table are mean (*n* = 18 for plant height, leaf length, SPAD, and RGR; *n* = 9 for Fv/Fm). Values in the same columns of each plant species with different lowercase alphabet letter are significantly different (*p* < 0.05). ns, *, **, *** mean no significant or significant at *p* ≤ 0.05, respectively. ^x^ Relative growth rate. - means no data.

**Table 6 foods-12-00069-t006:** Average nutrient content and nutrient use efficiency of red orache, minutina, and Swiss chard from the first harvest.

Plant Species	Treatment		N (%)		NUE ^x^	P (%)		PUE ^y^
	pH	Additional C	Shoot	Root		Shoot	Root	
Red orache	6.5	+C	3.19 b	2.69	14	0.91 a	0.85 a	2.4 a
		No C	3.55 b	2.40	13.4	0.60 b	0.48 b	1.3 b
	7.5	+C	3.28 b	2.28	11.4	0.62 b	0.46 b	1.2 b
		No C	4.80 a	-	9.2	0.59 b	-	0.7 b
	*p*		***	ns	ns	***	**	***
ANOVA								
pH			**	-	ns	***	-	***
Additional C			***	-	ns	***	-	***
pH × Additional C			*	-	ns	***	-	ns
Minutina	6.5	+C	2.42 b	2.23 c	10.5	0.76 ab	2.03 ab	2.1 ab
		No C	2.57 b	2.91 ab	12.8	0.61 ab	1.80 bc	1.7 ab
	7.5	+C	2.54 b	2.54 bc	12.0	0.92 a	2.25 a	2.5 a
		No C	3.39 a	3.45 a	13.3	0.57 b	1.51 c	1.3 b
	*p*		***	***	ns	*	**	*
ANOVA								
pH			**	**	ns	ns	ns	ns
Additional C			**	***	ns	*	***	*
pH × Additional C			*	ns	ns	ns	*	ns
Swiss chard	6.5	+C	3.14 b	2.73	16.7	1.31 a	1.35 a	4.1 a
		No C	3.63 ab	3.37	20.1	0.55 b	0.84 b	1.8 b
	7.5	+C	3.13 b	2.88	19.3	0.56 b	0.67 b	1.9 b
		No C	4.23 a	-	15.0	0.30 b	-	0.7 c
	*p*		**	ns	ns	***	**	***
ANOVA								
pH			ns	-	ns	***	-	***
Additional C			**	-	ns	***	-	***
pH × Additional C			ns	-	*	*	-	**

Values in the table are mean (*n* = 6). Values in the same columns of each plant species with different lowercase alphabet letter are significantly different (*p* < 0.05). ns, *, **, *** mean no significant or significant at *p* ≤ 0.05, respectively. ^x^ Nitrogen use efficiency. ^y^ Phosphorus use efficiency. - means no data.

**Table 7 foods-12-00069-t007:** Average nutrient content and nutrient use efficiency of okahijiki, kale, and Swiss chard from the second harvest.

Plant Species	Treatment		N (%)		NUE ^x^	P (%)		PUE ^y^
	pH	Additional C	Shoot	Root		Shoot	Root	
Okahijiki	6.5	+C	4.17	3.22	18.9 a	1.29 a	0.87	3.4 a
		No C	4.45	3.39	12.2 ab	1.03 b	0.65	1.6 b
	7.5	+C	4.88	3.29	8.3 bc	1.19 ab	0.67	1.2 bc
		No C	4.87	-	2.1 c	1.07 ab	-	0.3 c
	*p*		ns	ns	**	*	ns	***
ANOVA								
pH			ns	-	***	ns	-	***
Additional C			ns	-	**	*	-	**
pH × Additional C			ns	-	ns	ns	-	ns
Kale	6.5	+C	5.13 ab	3.28	23.9 a	0.75 a	0.79	2.0 a
		No C	5.20 a	3.46	23.3 a	0.74 a	0.69	1.9 a
	7.5	+C	4.63 bc	3.37	22.1 a	0.74 a	0.68	1.6 ab
		No C	4.40 c	3.52	12.3 b	0.57 b	0.66	1.0 b
	*p*		**	ns	**	***	ns	**
ANOVA								
pH			***	ns	**	**	ns	**
Additional C			ns	ns	*	*	ns	ns
pH × Additional C			ns	ns	*	*	ns	ns
Swiss chard	6.5	+C	4.58 a	3.86	30.0 ab	0.68 a	0.97 a	2.6 a
		No C	4.36 a	3.96	30.9 ab	0.61 a	0.74 ab	2.5 a
	7.5	+C	4.34 a	4.16	36.4 a	0.30 b	0.71 b	1.5 b
		No C	3.45 b	4.18	14.7 b	0.29 b	0.72 b	0.7 b
	*p*		***	ns	*	**	*	***
ANOVA								
pH			***	*	ns	***	*	***
Additional C			***	ns	*	ns	ns	ns
pH × Additional C			*	ns	*	ns	ns	ns

Values in the table are mean (*n* = 6). Values in the same columns of each plant species with different lowercase alphabet letter are significantly different (*p* < 0.05). ns, *, **, *** mean no significant or significant at *p* ≤ 0.05, respectively. ^x^ Nitrogen use efficiency. ^y^ Phosphorus use efficiency. - means no data.

**Table 8 foods-12-00069-t008:** Two-way ANOVA of total phenolic content and antioxidant capacity (ABTS and DPPH) of plants from first and second harvests.

Phenolics						
ANOVA	First Harvest			Second Harvest		
	Red Orache	Minutina	Swiss Chard	Okahijiki	Kale	Swiss Chard
pH	ns	*	***	***	***	*
Additional C	***	***	ns	***	*	***
pH × Additional C	***	ns	***	ns	ns	ns
Antioxidant capacity (ABTS)						
ANOVA	First Harvest			Second Harvest		
	Red orache	Minutina	Swiss chard	Okahijiki	Kale	Swiss chard
pH	***	*	**	***	ns	ns
Additional C	***	***	ns	ns	ns	***
pH × Additional C	***	ns	***	ns	**	ns
Antioxidant capacity (DPPH)						
ANOVA	First Harvest			Second Harvest		
	Red orache	Minutina	Swiss chard	Okahijiki	Kale	Swiss chard
pH	ns	ns	***	***	***	ns
Additional C	***	***	ns	***	ns	***
pH × Additional C	ns	ns	***	*	ns	ns

ns, *, **, *** means no significant or significant at *p* ≤ 0.05, 0.01, or 0.0001, respectively.

## Data Availability

The data presented in this study are available on request from the corresponding author. The data are not publicly available due to intellectual property policies.

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
