# Peer review of "Supplemental C Addressed the pH Conundrum in Sustainable Marine Aquaponic Food Production Systems"

_foods, 2022, doi:10.3390/foods12010069_

Round 1

Reviewer 1 Report

Environmental pH and supplemental C exert little impact on shrimp, but profound impacts on plants in marine aquaponics

Comments to authors:

General Comments:

The authors evaluate pH levels and additional C on the growth  and production of whiteleg shrimp and five plant species in experimental marine aquaponic systems. This work is fully integrated within the up to date and relevant understanding in sustainable food production. However, I would suggest performing modifications in order to improve the clarity of the study.

Specific Comments:

-Please, have a think about the title and whether it will be attractive to a wide audience - emphasise the fundamental science aspects of the work.

-Make sure the abstract starts with the main important findings of the study in order to increase the readers' interest.

-Introduction could be increased and restructured to better establish the main objectives of the study. Please ensure the hypotheses are clearly given and testable and that they relate to the aims and the objectives of the work.

-Results and Discussion sections should be better explored, increased and restructured to point out the main data. Many of the figures and tables do not add very much to the text, they could be deleted.

-Make sure the conclusions are supported by the data presented.

-Please, make sure the references are up to date and that you have checked recent issues of Foods

In conclusion, I hope the comments and suggestions above may be of helping to the authors for improving a version of the manuscript.

Reviewer 2 Report

The theme is attractive and promising. The introduction is pertinent, the text is well written, and the photos/tables clarify the interpretation of the work (A photo of shrimps would be very helpful).

Experiments seem convenient to me.

However, I think that the data has not been adequately treated, and the conclusions may not be correct.

Here I add some recommendations

0 RO (reverse osmosis) water is not a suitable water source for... (define abbreviation).

90 to adjust the salinity to15 ppt (add space)

.137 The amount of molasses added was determined on the carbon-nitrogen content of shrimp feed and the carbon content of the molasses to raise the...

How were determined the carbon content of molasses and the carbon-nitrogen content of shrimp feed?

180 Water samples were collected from the aquaculture tank twice a week before feeding, and were analyzed…

How many repetitions?

184 Alkalinity and total suspended solids (TSS) were measured once per week using HACH reaction kits and US EPA method, respectively.

How many repetitions?

188 Shrimp growth parameters such as initial weight, final weight, the number of shrimp, and the total feed input were collected at the beginning and end of the experiment ...

Shrimp weights were determined for each individual, or was the whole group weighed directly?

196 Plant growth parameters such as plant height (cm), leaf length (cm), and SPAD value (which indicates the content of chlorophyll per unit leaf area; SPAD-502 Chlorophyll meter; Minolta Camera Co. Ltd., Japan) were measured every two weeks

leaf length : which leaf? Or how many leaves per plant?

SPAD value: Was it taken from a particular leaf?

240 Statistical analysis

Shrimp and plant growth parameters, nutrient and antioxidant concentrations in plants, solid wastes, and water quality parameters were analyzed using JMP Pro16.0 (SAS 243 Institute Inc., Cary, NC) …

How many repetitions?

 245 If ANOVA indicated significant treatment effects, differences between means were determined by Tukey’s honestly significant difference test (HSD) at p ≤ 0.05. Add references e.g.  Tukey, John (1949). "Comparing Individual Means in the Analysis of Variance". Biometrics. 5 (2): 99–114.

259 Table 1.

I think it would facilitate the interpretation of the work if the initial and final number of shrimp per tank were added to the table.

Each value represents means±SD: 

What does SD indicate?: the dispersion of each individual with respect to the group mean, or the dispersion of what is harvested in each repetition with respect to the mean of the factor?

For example:

Table 1. Initial Weight (g) 2.25±0.00. I think that a shrimp population cannot have SD=0.

Others values ​​of SD make me doubt the assumption of homoscedasticity, consequently, the results of the ANOVA do not seem reliable to me.

I would reanalyze the data, considering the values ​​of each individual. Not just the repetitions.

I understand that for plants it was analyzed in this way.

286 Table 2. Fresh and dry weight of red orache, minutina, and Swiss chard in aquaponics under two 

add the Standard deviation 

Table 4 and 6

correct minutian by minutina.

312

Figure 2. P

There are morphological differences in minutina.

465The better growth, yield, and higher P concentration in + C treatments may be attributed to the addition of molasses, which provide a high concentration of C  to improve nutrient assimilation. The result of the present suggest that under saline conditions and/ or high environmental pH, providing external C could improve plant’s growth and yield.

there is a change in the font size

Round 2

Reviewer 1 Report

SUPPLEMENTAL C ADDRESED THE PH CONUNDRUM IS SUSTAINABLE MARINE AQUAPONIC FOOD PRODUCTION SYSTEMS

General Comments to authors:

The authors evaluate pH levels and additional C on the growth and production of whiteleg shrimp and five plant species in experimental marine aquaponic systems. They have made the major revisions I requested to the previous version of the manuscript, and this has greatly improved with respect to the original submission. Consequently, I suggest accepting the manuscript in its present form.

Author Response

Thank you for taking the time to review and provide suggestions for us to improve the manuscript.

Reviewer 2 Report

I thank the authors for having answered my questions so conscientiously.

However, it seems to me that the article should be worked on a little more in some concepts.

.

First

Some clarifications on methodology (e.g.  In order to reduce the stress, we measured 7 shrimp at a time and measured 4 times to… or ... We chose leaves that are fully expanded. Picked 3 leaves per plant and averaged the three data points for each plant) should be added to the paper.

570 Applying additional C was  is suggested as a solution of pH conundrum for the operation of marine aquaponic …

Last but not least:

564.  The current study found no significant effects of pH or additional C on shrimp performance.

It is difficult for me to accept this statement.

From Table 1.

Initial number

of shrimp per

tank

28±0

28±0

28±0

28±0

Final number

of shrimp per

tank

12±3

15±5

5±7

8±4

Survival rate (%)

42.8±9.5

54.7±18.0

19.0±23.8

27.4±12.5

It is evident that the Survival rate drops 50% or more when the pH is 7.5 (additional C slightly attenuates this drop).

Regarding weight gain (WG) and specific growth rate (SGR)

I remind the response of the authors:

 In order to reduce the stress, we measured 7 shrimp at a time and measured 4 times to

get 28 shrimp per tank instead of measuring individually. This means we have 4 data points per

tank (treatment repetitions).

1) this sentence should be added to the paper

2) I do not understand how this procedure could be executed on the final stage when the number of shrimps was reduced to half (in the best of cases).

I think it is necessary to clarify more how these variables were determined.
